# Beer Molecules and Its Sensory and Biological Properties: A Review

**DOI:** 10.3390/molecules24081568

**Published:** 2019-04-20

**Authors:** Bruno Vieira Humia, Klebson Silva Santos, Andriele Mendonça Barbosa, Monize Sawata, Marcelo da Costa Mendonça, Francine Ferreira Padilha

**Affiliations:** 1Biomaterials Laboratory (LBMat), Institute of Technology and Research (ITP), Av. Murilo Dantas, 300, Aracaju 49032-490, Sergipe, Brazil; andrielemendonca@yahoo.com.br (A.M.B.); monisawata@gmail.com (M.S.); marcelo_costa@unit.br (M.d.C.M.); francinefpadilha@gmail.com (F.F.P.); 2Tiradentes University (UNIT), Av. Murilo Dantas, 300, Aracaju 49032-490, Sergipe, Brazil; 3Center for Study on Colloidal Systems (NUESC)/Institute of Technology and Research (ITP), Av. Murilo Dantas, 300, Aracaju 49032-490, Sergipe, Brazil; 4Empresa Brasileira de Pesquisa Agropecuária (EMBRAPA), Avenida Beira-mar, 3.250, Aracaju 49025-040, Sergipe, Brazil

**Keywords:** volatile esters, phenolic compounds, beer, alcoholic fermentation, beer adjunct

## Abstract

The production and consumption of beer plays a significant role in the social, political, and economic activities of many societies. During brewing fermentation step, many volatile and phenolic compounds are produced. They bring several organoleptic characteristics to beer and also provide an identity for regional producers. In this review, the beer compounds synthesis, and their role in the chemical and sensory properties of craft beers, and potential health benefits are described. This review also describes the importance of fermentation for the brewing process, since alcohol and many volatile esters are produced and metabolized in this step, thus requiring strict control. Phenolic compounds are also present in beer and are important for human health since it was proved that many of them have antitumor and antioxidant activities, which provides valuable data for moderate dietary beer inclusion studies.

## 1. Introduction

Beer consumption is widespread since it is the most consumed alcoholic beverage, and the third amongst general beverages. In ancient times, beer was widely used for human nutrition, in religious practices and also for disease treatments. Currently, there are many beer styles and types, depending on the brewing process and ingredients used. The addition of hops improved the taste and the protection of beer due to its pH lowering effect and antibacterial activity, which inhibits Gram-positive bacteria, but it is ineffective against brewer’s yeast, making beer a safe drinking source [1,2,3,4].

The definition of a beer pattern style is based on many factors, however, it is the apparently generic fermentation step itself that provides the base from which the most recognizable organoleptic characteristics will arise. Brewery fermentation is mainly described as the energetic yeast metabolism of a fermentable carbohydrate source in the absence of oxygen, resulting in alcohol and carbon dioxide production. The brewery fermentation can be carried out in high or low temperatures. Lager beer is the most widespread style and it is primarily produced with *Saccharomyces pastorianus* yeast strains, being fermented at temperatures that range from 3.3 to 13.0 °C for 4–12 weeks. In contrast, ale beers, which are more prevalent in northern countries, such as Germany, Belgium, Canada, and Britain, are typically fermented at higher temperatures ranging from 16 to 24 °C for 7–10 days, by the top yeast strain, *Saccharomyces cerevisiae* [1,5,6,7,8].

The beer aroma is known to be derived primarily from innate chemical volatile compounds of the barley malt (or as a result of thermal treatment during malting), hops and yeast metabolism (development of beer during fermentation and aging). Currently, several different volatile compounds that can affect the final flavor quality of beer have been identified. They can be divided into five groups: (i) from ingredients, such as barley malt and hops, (ii) from roasting malt and boiling wort, (iii) as yeast metabolism by-products during fermentation, (iv) from microorganism contamination, (v) from inappropriate storage conditions, such as oxygen and sunlight exposure. The volatile compounds affect beer’s organoleptic profile and are composed mainly of aliphatic and aromatic alcohols, esters, organic acids, aldehyde, carbonyl compounds, and terpenic substances. Although the raw materials are practically uniform to all beer styles, some aromas and flavors are unique regarding traditionally produced beers and appear to be related to yeast strains’ metabolism during aging. Many studies are being conducted regarding the chemistry of beer aroma compounds, especially in terms of the composition and structure of volatile esters, which can greatly vary between different traditional brewing processes [7,8,9,10,11,12,13,14,15,16,17].

There are several methods concerning the identification and quantification of complex, volatile organic compound signatures of beer headspace, however, these approaches could be improved in order to achieve more complete chemical information. For modern brewing technology, a better understanding of the key volatile aroma compounds is of primordial importance, which optimizes the raw materials selection process and the yeast strain choice, as well as for quality control protocols. Therefore, in addition to the socio-cultural aspects related to beer consumption, the aim of this review is to describe several beer bio-compounds, recognize their nutritional function and their role for beer sensory attributes [12,15,16,17].

## 2. The Brewing Process

Beer is an alcoholic beverage produced as a result of a sugar wort fermentation process. Beer is derived from malted cereals and grains, most commonly barley and wheat, and less commonly from sorghum, starchy vegetables, and rye, along with water, hops and a yeast strain. Malting is the first step of brewing and consists of barley (or other cereals) controlled germination at lower temperatures. Malting might run at a lower temperature in order to minimize respiratory loss of carbohydrates, rootlet growth, and allow grains germination. Germination activates sugar degradative enzymes, such as α-amylase, amyloglucosidase and β-amylase, which further hydrolyze the clustered stored starch into fermentable sugars that are used by yeast’s energetic metabolism. The germinated malt grain is then carefully roasted to dry it to cease germination but also to allow the maintenance of the enzymes’ degradative ability [3,17,18,19,20,21,22].

The resulting malt is subsequently milled to grist and added to the mashing vessel with hot water and kept at a temperature of approximately 62 °C (amylase rest) to start the mashing step. At this point, the starch granules swell and allow its conversion into fermentable sugars by enzymes including α- and β-amylase, starch de-branching enzyme, and α-glucosidase. Additional temperature steps are programmed to allow, for instance, other enzymatic activity to proceed in the mashing process, as phytase (pH lowering), proteases and peptidases (proteins hydrolysis). Starch is hydrolyzed to oligosaccharides with up to four polymerization degrees (DP4), as maltose, maltotriose, fructose, glucose and sucrose [3,18,23,24,25].

At the end of the mashing step, a sugar-containing liquid, or wort, is formed and brought to boiling, when the hops and/or spices are added. Hops are conical shape flowers of the *Humulus lupulus* vine, which give beer its pattern, bitter flavor, and aroma. With the increasing heating during boiling, a bulk of proteins is precipitated forming a curd and a dense foam. This enhances the wort with a particular group of water-soluble and heat stable protein families formed mostly of lipid transferase proteins and serpin Z4. The high temperature provided by boiling allows hops α-acids humulones isomerization into trans-isohumulones, which is the liable bitterness fraction. Following, a filtration system through a bed formed by the spent grain husks (lautering) is conducted to extract the particulate plant matter, precipitated proteins and debris. The wort is then briefly cooled to a yeast-compatible growing temperature and oxygenated [4,26,27,28,29,30].

The oxygenated and cooled wort is fermented in the presence of the selected yeast strain, and sugars are converted into alcohol and carbon dioxide. The final stage of brewing is the maturation, in which beer is stored at lower temperatures, depending on the style of beer, for several weeks. In large industries, beers are then filtered and pasteurized to remove the yeast and stabilize the beer prior to packaging. However, most craft beers and historic local beers, e.g., traditional Bavarian *weissbier*, are not filtered [1,3,7,31,32,33].

German beer purity law was established to avoid hazardous raw materials from being used, such as rushes, roots, spices and in some cases animal derivatives. However, wheat, rice, rye, oats, maize, unmalted barley, and to a lesser extent sorghum, millet and cassava have all been used in brewing. Most of these grains, particularly rice and maize, are often used by great industries as adjuncts, in order to supplement the primary mash ingredient, and produce a more cost-effective product. Wheat and oat are commonly used as adjuncts due to their ability to promote foam stability, and prior to the use of hops, other bitter herbs, spices, and flowers added in order to bring different sensory profiles and to create personalized beers [8,17,21,24,34,35,36,37].

## 3. Beer Volatile Esters

Fermented beverages, mainly beer and wine, contain only traces of esters as volatile compounds, however, they have pronounced importance for the aroma and flavor profile of grid beverages. The most well-described flavor-active esters in beer are ethyl acetate (solvent-buttery like aroma), ethyl caproate, ethyl caprylate (sour apple-like flavor and aroma), isoamyl acetate (fruity, banana aroma), isobutyl acetate, phenylethyl acetate, and ethyl octanoate (honey, fruity, roses, flowery aroma) [9,10,38,39].

Aroma-active esters are synthesized by fermenting yeast cells in the intracellular space. It has been demonstrated that the esters diffuse amongst cells and fermenting medium depending on the yeast species used and the temperature since most esters are retained at lower temperatures. The higher proportion of esters produced remains inside cells of lager yeasts (*Saccharomyces pastorianus*, *Saccharomyces carlsbergensis*, *Saccharomyces uvarum*), and in the fermenting medium for ale yeasts (*Saccharomyces cerevisiae*), which explains the higher complex flavor combination of ale beers [7,39,40,41,42,43].

Acetate esters’ aroma is promptly sensory perceived since they rapidly diffuse through the yeast cellular membrane into the wort-fermenting medium due to their lipid-soluble characteristic. The ratio of medium-chain fatty acids ethyl esters (FAEE) transferred to the medium, in contrast to acetate esters, decreases with the chain length increasing: 99% for ethyl caproate, 52–69% for ethyl caprylate, and 7–19% for ethyl caprate. Longer chain fatty acid ethyl esters remain entirely in the intracellular environment [9,38,44,45].

It was also perceived, primarily by sensory analysts, and subsequently elucidated by gas chromatography (GC) quantification assays, that esters may affect beer flavor below their individual threshold concentrations. It has been described that the presence of varied esters can present synergistic effects on individual flavors and interfere with the whole traditional profile of beer. Moreover, since most esters are present in concentrations ranging in the threshold value, minor variations in their concentration may have critical effects on beer’s organoleptic properties. It is well established that brewing with high gravity worts results in a significant acetate esters overproduction, thus in order to achieve more monitoring over ester synthesis and revert the adverse implications, a great deal of research has been devoted to clarifying the yeast ester metabolism and its biochemical background [39,44,45,46,47].

Volatile aroma ester synthesis occurs in the intracellular environment of the yeast, and these esters are the final products of an enzyme-catalyzed condensation reaction between acyl-CoA and higher alcohols. Several enzymes with different structures and activities are associated with ester production. However, the alcohol acetyl transferases I and II (AATase I and II), which are respectively encoded by ATFl and ATFZ genes, have a well-established pathway. Other well-known enzymes involved in cellular ester synthesis are Lg-Atf1p, an AATase which are found primarily in lager yeast, and the ethanol hexanoyl transferase (Eht1p), an enzyme associated with the catalytic formation of ethyl hexanoate. The Atf1p that is homologous to Lg-Atf1p and Atf2p enzymes and are partially related to isoamyl acetate and ethyl acetate production. Moreover, previous studies have shown that the balance between esterases (enzymes that hydrolyze esters), such as Iah1p, and ester-synthesizing enzymes, may be critical for the final ester concentration in finished beer. This balance is achieved during beer aging since there is a decrease in yeast metabolism due to a lowering in the internal temperature, and an increased rate of ester resorption by the yeast itself [10,38,39,45,48,49,50,51,52].

Several parameters, used in the malt production, can also affect the final flavor and aroma of beer, with an emphasis on dark malts, which are used to brew certain beer styles. This sensory profile is mainly a result of the Maillard reaction products (MRPs), which are synthesized due to the high temperatures used in dark malts production. Sensory and GC–MS analysis studies show that the processing baseline used in dark malt production widely interferes in the Maillard reaction-derived aroma compounds present in wort and, consequently, the beer flavor profile [10,12,14,44].

Studies show that esters were found to decrease with the use of dark malts, even though the presence of higher alcohols provides the substrate for acetate ester synthesis. Since substrate availability was not a limiting factor for the decrease in ester synthesis, it is likely to assume that the metal-chelating characteristics of MRPs, notably melanoidins, may cause an overall decline in the enzymatic activity, likely through chelation of cofactors, such as magnesium. Magnesium is a key cofactor during the brewing process because it acts in a large number of catalytic reactions and plays a major role in yeast cells protection against environmental stresses, such as the high ethanol concentration in fermenting wort. Amongst its many functions, magnesium acts as a cofactor for phosphoglycerate kinase in the synthesis of pyruvate and ATP and is directly involved in the reduction of fusel alcohols to fusel acids, since it activates the cytosolic aldehyde dehydrogenase ALD6 enzyme [10,39,45,49,53,54].

Studies also have shown that other co-substrate availability, such as higher alcohols, might be the major limiting factor for ester production. It was demonstrated that 3-methyl-1-butanol supplementations to both, normal and high-gravity worts, leads to increased production of isoamyl acetate. Other studies carried out with yeast transformant cells also demonstrated that cells overexpressing the cytosolic branched-chain aminoacid aminotransferase-encoding gene, BAT2, are able to produce at least 1.3 times more isoamyl alcohol and 1.5 times more isoamyl acetate. Therefore, it has been shown that mutant yeast cells and transformants that overproduce specific higher alcohol, will exhibit a corresponding increase in the respective acetate ester synthesis. This suggests that the fusel alcohol availability actually impacts the production of the derived esters [9,55,56,57].

The residual esterase activity in finished beer may also lead to a decreasing ester concentration as is often observed during long storage periods. Basically, it is well established that the two factors that are of great relevance for ester production rate: the acyl-CoA and fusel alcohol concentrations, and the total availability and suitability of the enzymes involved in the formation and cell release of the respective ester. Therefore, all parameters that influence substrate availability or enzyme activity will affect the final ester concentration [38,39,46].

Figure 1 shows a model for ester production during brewing fermentations related to acetyl-CoA substrate availability as the key limiting factor. Controlled parameters, such as fatty acid addition, nitrogen and oxygens levels, nutrients availability, temperature, and pressure are directly involved in ester synthesis since they might alter the levels of acetyl-CoA in the fermenting wort. In brief, every variable that modifies acetyl-CoA levels would also change ester production and concentration. The high levels of wort solids, wort lipids, and oxygen also interfere, since they induce yeast growth and consequently the acetyl-CoA concentration, which decreases acetyl-CoA availability for ester production. However, there are few studies concerning the influence of glucose or nitrogen addition and the decrease of top pressure, which demonstrably increases both, yeast growth and ester production [14,42,48,51]. 

Another parameter that affects ester production is mycotoxin contamination, which may occur at different stages of brewing. Many of them may be transferred from cereal grains to malt and then to beer due to their high-temperature resistance (aflatoxins, zearalenone, and deoxynivalenol) and water solubility (deoxynivalenol and fumonisins). Studies concerning the effect of mycotoxin contamination of wort on alcoholic fermentation volatile compounds found that some mycotoxins (mainly AFB1 and deoxynivalenol) are capable of inhibiting alcohol dehydrogenase. The result is an increase in acetaldehyde concentration and other undesirable volatile compounds synthesized during alcoholic fermentation, but it has no effect on the total ester content [58,59,60].

It is well established, however, higher alcohol availability in the fermenting wort itself does not explain the effects of some parameters on ester synthesis. Studies have shown that fusel alcohol production is increased by the presence of high levels of oxygen and unsaturated fatty acids, but they lead a decrease in ester content. A third suggested model was developed by placing the AATase enzyme in a central role. This model showed that the enzymatic activity follows a similar behavior to that of ester synthesis and it is repressed by high oxygen content and linoleic acid supplementation to the medium. It has been suggested that this regulation is mediated by the Rox pathway [10,40,49,61].

Studies have also demonstrated that a low-oxygen response element (LORE) co-regulates the response of ATFl and the A9 fatty acid desaturase-encoding gene, OLEl. It has been shown this promoter element is selectively repressed by the unsaturated fatty acids concentration and its activation occurs under hypoxic condition. Furthermore, previous studies have demonstrated that ATF1 activity is regulated by protein kinase A (PKA) and Sch9p kinases proteins. These enzymes play a significant role in the transcriptional gene regulation in response to changes in substrate concentration, such as carbon, phosphate, and nitrogen. The main targets of Sch9p are genes related to cell growth, stress response, and the metabolism of trehalose and glycogen. In addition, studies carried out by Gallone et al. (2016) suggest that ester synthesis in wort fermentation is not restricted by substrates’ availability, supporting the key role of AATase activity, since it has been demonstrated that brewer’s yeast strains genetically modifies overexpressing. The ATF genes produce at least five times more isoamyl acetate and ethyl acetate in fermented wort [9,14,48,62].

As AATase activity is suggested to be restricted mostly to ATF gene expression, the real limiting factor for acetate ester synthesis in yeast cells is the transcriptional activity of the ATF genes. However, in addition to ATF gene expression, the effects of substrate concentrations on ester production should not be discharged. Several variables influence ester production and there is an opportunity for brewers to control the beer ester content. Therefore, many settings are related to both the regulation of AATase activity (multiple regulation mechanisms at ATF expression level) and in substrate availability regulation (carbon, nitrogen, oxygen, and fatty acid metabolism), even though recent research has brought some advances to the brewing industry. The control of ester formation is extremely complex and difficult to predict. Table 1 shows the main compounds responsible for beer sensory characteristics described in the literature and their detection threshold [39,40,41,62,63].

## 4. Complementary Beer Volatile Compounds

Beside esters, the other well-described volatile compounds are structurally classified into higher alcohols, and carbonyl compounds, such as ketones and aldehydes, and sulfur-containing compounds [33,60].

### 4.1. Higher Alcohols

The bottom-fermenting yeast (*Saccharomyces carlsbergensis*) is generally used in regular beer fermentation processes and in low-malt and low-alcohol beer fermentation. These yeast strains have been reclassified as *Saccharomyces pastorianus*, a natural hybrid form between *Saccharomyces cerevisiae* and *Saccharomyces bayanus* and is often used in lager beer production. Therefore, bottom-fermenting lager beer yeast includes two genomes portions: *Saccharomyces cerevisiae*, (Sc)-type, and lager, (Lg)-type. However, to investigate gene functions related to volatile compounds synthesis, many authors and researches often use *Saccharomyces cerevisiae*, because it has a wide experimental background and is easier to manipulate in biotechnology and genomic technology. In this review, the research papers have focused on the discussed results from the use of *Saccharomyces cerevisiae* [9,42,43,50,62].

Amyl alcohol is reported to be the most present and quantitatively significant flavor compound of higher alcohol groups. Active amyl and its isomer isoamyl alcohols are, most of the time, described as represented purely as amyl alcohol. This compound affects beer drinkability since beer flavor is described by sensory analysis as heavier when amyl alcohol content increases. Another higher alcohol that affects beer quality is isobutyl alcohol, and its undesirable effect can be perceived when its concentration in beer exceeds 20% of the total concentration of three other alcohols, such as *N*-propanol, isobutyl, and amyl [9,10,61,64].

There are two well-known metabolic pathways for higher alcohol biosynthesis in *Saccharomyces cerevisiae* (Figure 2), one is from glycolysis and the other from aminoacids. In both pathways, α-keto acids are formed, decarboxylated to aldehydes, and dehydrogenated to produce the corresponding primary alcohol. In the Ehrlich pathway or the aminoacids pathway, the aminoacids are converted to α-keto acids by the aminotransferase enzyme. Esters are synthesized from higher alcohols and acyl-CoA, and their production is closely connected to higher alcohol precursor production. Thus, it is important for brewers to elucidate higher alcohol synthesis and the mechanisms of higher alcohols to esters conversion, during the research of alcoholic beverage volatile compounds and flavors. Several studies were carried out in order to examine the production of higher alcohols and they have employed many recombinant yeast strains, which were modified based on the information derived from their specific metabolic pathways [41,65,66,67,68,69]. 

Toh et al. (2018) studies focused on the volatile compounds production by simultaneous fermentation with the yeast species *Saccharomyces cerevisiae* and *Torulaspora delbrueckii*, and demonstrated that amongst the volatile compounds, alcohols made up the largest relative proportion. It was observed that for isobutanol, an Ehrlich pathway by-product of valine, all beers yielded no significant difference in their concentration. However, for the fermentation with mixed cultures, the 1:20 (*Saccharomyces cerevisiae*:*Torulaspora delbrueckii*) inoculum ratio of the yeast strains seemed to be promising in terms of fusel alcohol production compared to the *Saccharomyces* monoculture, with considerably more isoamyl alcohol and similar levels of 2-phenylethyl alcohol. This could be because isoamyl alcohol was more abundantly available and diverted toward esterification for isoamyl acetate generation by *Saccharomyces cerevisiae* strains. The result is a markedly lower isoamyl alcohol level but higher isoamyl acetate concentration. These results show how both, esters and higher alcohols, biosynthetic pathways can be interchangeable [13].

A study carried out by Schoodermark-Stolk et al. (2005), has discussed the effects of the suppression BAT1 and BAT2 genes, which are responsible for encoding a branched-chain aminoacid aminotransferase enzyme in *Saccharomyces cerevisiae*. The study has demonstrated that BAT2 single- and BAT1-BAT2 double-suppression mutants, grown on glucose presence, can produce isoamyl alcohol. Although, these yeast strains were unable to synthesize isoamyl alcohol in the presence of ethanol alone as a single carbon source. It was suggested that BAT2 expression is necessary for isoamyl alcohol formation in an ethanol-high medium since BAT1 is essential for isoamyl alcohol biosynthesis in the glucose-enriched medium since from gene expression analysis, in which BAT2 was enhanced, the yeast mutant strain was grown in ethanol as the only carbon source. BAT2 overexpression was also examined and resulted in 1.3-fold and 2.2-fold increases in isoamyl alcohol and isobutyl alcohol biosynthesis, respectively [70]. 

Another study focused on the ECA39 and ECA40 genes in *Shinzosaccharomyces pombe* (similar corresponding genes BAT1 and BAT2 in *Saccharomyces cerevisiae*). It has been reported that these genes encode cytosolic and mitochondrial branched-chain amino acid (BCAA) aminotransferases. It was demonstrated by using *Saccharomyces cerevisiae* mutants, in which the genes encoding BCAA aminotransferase were suppressed, that these modifications had no effect on n-propanol production. However, the ECA40 mutant demonstrated a significant reduction in isobutyl alcohol production, and surprisingly higher isoamyl alcohol concentrations were noted in the ECA39/ECA40 double-suppression mutant [9,71,72].

### 4.2. Carbonyl Compounds

The concentration of carbonyl compounds in beer is low when compared with other volatile by-products. Even acetaldehyde, which is the predominant carbonyl compound in beer, is usually present at no more than 10 ppm, but it is enough to affect beer drinkability. Diacetyl is a carbonyl compound produced during primary fermentation, and its concentration is usually used as an indicator of wort fermentation or maturation quality. Diacetyl concentration should not exceed 0.1 ppm, since its threshold from 0.05–0.1 ppm and higher concentrations may develop a stale milk aroma and flavor, and it is often reported as an undesirable volatile compound. A sulfur-containing volatile compound, 3-methylbut-2-ene-1-thiol (MBT), which synthesis pathway has been previously elucidated by researchers, is produced not by yeast metabolism during fermentation, but by the degrading effects of the light on light-sensitive compounds of the beer, and gives off-flavors described as skunk-like odor at concentrations of a few tens of ng·L^−1^ [68,73,74].

Studies concerning the potential reducing of acetaldehyde concentration, existing as an intermediate to ethanol production, investigated a mutant yeast strain with a disrupted alcohol dehydrogenase II gene (ADH2). It has been shown that ADH1 encodes a protein that performs as the major catalyst of acetaldehyde reaction, by contrast with ADH2, which predominantly encodes a protein for ethanol intake and ethanol transfer to acetaldehyde. It was also demonstrated that, although there is no distinction in cell growth and concentrations of other volatile compounds during fermentation between the mutant and the pattern brewing yeast strains, the final acetaldehyde content from the mutant strain decreased to approximately one-third that of the pattern strain, which is an important advance to avoid or reduce acetaldehyde off-flavors in beer [7,11,53,73,75].

Diacetyl (2,3-butanedione), as mentioned before, is an important volatile compound that affects beer quality. It is an organic compound that arises naturally as a by-product of fermentation. In some fermentative bacteria, it is formed via-thiamine pyrophosphate-mediated condensation of pyruvate and acetyl-CoA. For beer, in terms of flavor, diacetyl is an important compound, however, its overproduction may affect the final quality of beer. At high concentrations (>0.1 ppm) diacetyl may bring buttery/solvent like flavor and a rancid mouthfeel to the beer. Although for some beer styles, like stouts, scotch ales and pilsners, diacetyl concentration slightly above the threshold is accepted, and it brings a toffee-like described flavor. For lagers, it is quite important to remove diacetyl, which is produced during the metabolism of valine from α-acetolactate by the yeast cells. α-acetolactate is spontaneously decarboxylated to diacetyl outside the cell during the maturation step, when it is converted into acetoin (3-hydroxybutanone), which has no sensory characteristics (Figure 3). However, during low maturation temperatures, this conversion is very slow [31,32,33,73,74,75,76].

The metabolism of sensory active compounds was most frequently modified to decrease the diacetyl content in beer. In previous studies, the main strategy for suppression of diacetyl formation was to increase the activity of α- acetolactate decarboxylase, the enzyme catalyzing the degradation of α-acetolactate (diacetyl precursor) to 3-hydroxybutanone. At first, Sone et al. (1988) studied the insertion of a DNA fragment containing the encoding gene of α-acetolactate decarboxylase from *Enterobacter aerogenes*, under the control of the alcohol dehydrogenase promoter, into the *Saccharomyces cerevisiae*. Laboratory-scale fermentation tests revealed that the mutants were able to lower the α-acetolactate concentration in the medium more intensively in comparison to the wild type strains. A similar approach has been adopted by several research groups [9,39,74,77].

In a more recent study, Cejnar et al. (2016) discussed the development of an engineered brewery yeast modified with the acetolactate decarboxylase encoding gene from *Acetobacter acetisspxylinum* anchored onto the surface of the cell wall. As a result of the activity of these yeasts, the highest diacetyl content was noted during the fermentation tests. It was approximately 30% lower compared with control yeasts. However, the tolerance of consumers to recombinant yeasts containing bacterial genes is still very low, which strengthens the need for new strategies focusing on the engineering of the original yeast’s metabolic pathways, enhancing valine biosynthesis from α-acetolactate [78].

An alternative way to reduce diacetyl concentrations was presented by Lu et al. (2012). It used an engineered yeast strain capable of metabolizing α-acetolactate in the cytosol before it leaves the cell into the fermenting wort. The mutant strain contained the acetohydroxy acid reductoisomerase (Ilv5pΔ46) encoding gene, whose enzyme was expressed and sited in the cytosol. In another recent work, carried out by Shi et al. (2016), three recombinant yeast strains with a disrupted acetolactate synthase (ILV2) gene were studied. They overexpressed 2,3-butanediol dehydrogenase (BDH1) gene and a combination of both. The use of all strains in fermentation tests showed a decrease of diacetyl content in comparison with the pattern strain. Shi et al. (2017), using an engineered yeast with overexpressing BDH2 and ILV5 genes and a disrupted ILV2 gene, showed that each of these strategies reduced the diacetyl content in fermenting tests. However, superior results have been obtained using recombinant strains with a deletion of ILV2 and simultaneously overexpressing ILV5. This research regarding flavor volatile compounds biosynthesis pathways might be relevant for future beer off-flavor control studies or to identify bacterial spoilage during the brewing process [79,80,81].

## 5. Beer Phenolic Compounds Profile

The phenolic compounds profile may directly influence beer’s sensory characteristics by adding an astringent and bitter flavor to the beverage. The types and concentration of phenolic compounds present in beer depend on the beer style and are related to the raw materials composition used in the brewing process. Such raw materials can present varied patterns according to the genetic characteristics of the source (grains, roots, hops), and the environmental conditions during cultivation. Moreover, the brewing step can also interfere with the final concentration of phenolics. As previously discussed, large breweries often use alternative means to produce a more cost-effective product, such as less expensive raw materials. In general, craft breweries only use raw materials, such as barley and hops, and less often, wheat during beer production, which may be related (in part) to the presence of different and more abundant phenolic compounds in craft beers [82,83,84].

Phenolic compounds are secondary metabolites of plants and may also contribute to beer color and aroma. These compounds have shown an antioxidant capacity and may be related to the oxidative stability of beer, although the main bitterness source originates from hops α-acids. Antioxidants often exhibit beneficial biological effects on human health, such as a reduction in cardiovascular disease and in oxidative damage to biomolecules and cells that prevent certain types of cancer. However, clinical trials and in vivo studies are still needed to confirm that moderate beer consumption is beneficial for human health [84,85,86].

Bettenhausen et al. (2019) study shows that during the germination phase of malting natural phenolic and antioxidant activities decrease, but then they increase considerably during steeping and kilning. Phenolic compounds, such as hydroxycinnamic acids (*p*-coumaric acid and isoferulic acid) and flavan-3-ols (catechins and epicatechins), in malt and beer have a strong impact on antioxidant activity (increased shelf-life) and colloidal stability (foam and haze) of beer. Polyphenolic compounds in beer are mainly derived from malt (75%) and hop (25%) and they are claimed to be one of the major sources of beer antioxidants [80,83,87]. Cheiran et al. (2019) studies identified fifty-seven phenolic compounds by HPLC-DAD-ESI-MS/MS measurements, and 12 of these compounds had never been previously identified in beer [82].

The mostly reported phenolic compounds in beer include flavonoids, phenolic acids, tannins, proanthocyanidins, and amino phenolic compounds, all of which have been described by their significant antioxidant activity, as well as other biological effects. Zhao et al.’s (2010) study determined the phenolic compounds profile of different commercial beers and showed (−)-epicatechin, gallic acid, protocatechuic acid, (+)-catechin, caffeic acid, vanillic acid, ferulic acid, *p*-coumaric acid, and syringic acid as the most common phenolics identified among beer samples. Gallic and ferulic acids were >50% of the total content of individual phenolic compounds found during beer studies and are the most reported phenolics in beer [88,89,90].

Different studies reported a relatively high content of vanillic and *p*-coumaric acids, and (+)-catechin while the values were lower for syringic acid and (−)-epicatechin in beer samples. Considerable variations can also be observed in phenolic profiles among different beer samples. It is assumed that the great variations in phenolic profiles for different beers might be due to differences in the raw materials, brewing process and original wort gravity. Previous reports that used Pearson’s product moment correlation coefficients, show that (+)-catechin and ferulic acid were the most efficient antioxidants in beer and the decrease or increase in antioxidant activity during brewing was accompanied by changes in the levels of (+)-catechin and ferulic acid. These phenolic compounds exhibited strong positive correlations with antioxidant activity assays (DPPH radical scavenging activity, ABTS radical cation scavenging activity and reducing power) [86,89,90,91,92].

It is difficult to isolate and characterize every compound in beer, and then evaluate their antioxidant activities due to the diversity and complexity of the natural antioxidants. Compounds with flavonoid structure, like (+)-catechin, generally show higher antioxidant activity than non-flavonoid compounds, such as phenolic acids, stilbenes, lignans, and coumarins. The activity of flavonoids to act as antioxidants depends upon their molecular structure, the position and number of hydroxyl groups and double bonds in the chemical structure. Flavonoids that possess multiple hydroxyl groups, especially 30–40 *o*-dihydroxy groups, and the 3- and 5-OH groups with 4-oxo function in A and C rings are generally more efficient antioxidants than non-flavonoid compounds. Indeed, as mentioned above, (+)-catechin has been found to have higher antioxidant activity than caffeic acid, ferulic acid, chlorogenic acid and *p*-coumaric acid assessed by the ABTS method. Therefore, it is efficient to improve beer antioxidant activity by increasing the levels of phenolic compounds, particularly flavonoids and some phenolic acids in beer [86,90,93,94].

New phenolic compounds in beer were recently described by Cheiran et al. (2019). Compounds derived from benzoic acid like 2,4-dihydroxybenzoic, 2,3-dihydroxybenzoic, and dimethoxybenzoic acids have not yet been reported. It is suggested that the origin of some of these compounds in beer is derived from barley, since the presence of 2,4-dihydroxybenzoic acid and dimethoxybenzoic acid in barley malt have been reported by Andersson et al. (2008), and Kim et al. (2007), respectively. Moreover, other isomers of dihydroxybenzoic acid have been identified in wheat, barley, rye, and oat. These compounds may prevent oxidative reactions in the beer itself and may also influence the sensory characteristics of astringency and/or bitterness [82,94,95,96,97,98].

4-*p*-Coumaroylquinic acid and 5-feruloylquinic were phenolic compounds that have been identified in hop bracts (i.e., leaf-like structures associated with hop cones) by Tanaka et al. (2014), and recently reported in beer. Breweries frequently use hops in pellets form, normally made from the whole hop, which ensures no part of hops flowers is removed. Thus, it is suggested that these compounds may be more concentrated in the bracts of the hops. These compounds and the derivatives of benzoic acid (cited above) play a beneficial role in health due to their antioxidant activities quercetin dihexoside and taxifolin hexoside (dihydroquercetinhexoside) were also recently described in beer samples and are also originated from hops. In beer, these flavonoids can play an antioxidant role, contributing to bitterness and to turbidity caused by binding barley proteins. Previous studies have reported that quercetin can act as an anti-inflammatory, antioxidant antiproliferative, aside from contributing to the mitigation of atherosclerosis [82,84,98,99,100].

## 6. Beer Compounds and Human Health

Many effects of beer compounds on biological systems have received special attention from the scientific community. However, while the harmful effects associated with the high intake of alcohol are well described, the effects of moderate doses are more complex to discuss and require further study. The issue remains on the possibly different effects of diverse alcoholic beverages in relation to their heterogeneous content of non-alcoholic components. It has been demonstrated that bio-compounds present in beer, such as flavonoids and other phenolic compounds, dramatically reduce the risk of developing cardiovascular disease [101,102], cancer [69,85], and have antioxidant protective effects [92]. However, beer consumption at high concentrations is not recommended for human health and it may lead to harmful effects in pregnant women, children, individuals with a clinical picture of cardiomyopathy, depression, pancreatic, liver and renal diseases, cardiac arrhythmias, and people at risk to develop alcoholism [15,25,26,27,28,102].

### 6.1. Antitumor Properties and Cancer Chemopreventive Potential Subsection

Several hundred beer constituents have been identified to date, that makes beer an extremely complex beverage. Total phenol content of 500–1000 mg/L have already been quantified in beer samples and a vast bulk of flavonoid polyphenols in most beers are described to be bound to polypeptides as soluble complexes. Most of the phenolic isolated compounds tested in various enzyme- and cell culture assays for their cancer chemopreventive potential showed that they play a significant role due to their antioxidant effects, modulation of carcinogen metabolism and anti-inflammatory mechanisms. Several structural classes of compounds have considerable distinct profiles. The catechins have been additionally identified as good Cox-1 inhibitors but only had marginal effects on Cyp1A and quinone oxidoreductase activity. Flavanones, on the other hand, were described as potent inhibitors of Cyp1A at nanomolar concentrations, suggesting their concomitant anti-inflammatory potential [85,92,101,103].

Previous studies showed that xanthohumol, a prenylated chalcone derived from the hop, is converted into the corresponding isomeric prenylflavanone isoxanthohumol during the brewing process. Xanthohumol was reported as a potential inhibitor of phase 1 Cyp1A activity, which indicates important chemopreventive behavior in the initial phase of carcinogenesis. Xanthohumol was also identified as a monofunctional inducer of NADH:quinone reductase activity, which affects the cDNA-expressed human Cyp1A1, Cyp1B1, Cyp1A2, Cyp3A4, and Cyp2E1 and inhibited the 7-ethoxyresorufin O-deethylase (EROD) activity of human Cyp1A1. The flavanones were described as more effective inhibitors of Cyp1A2 acetanilide 4-hydroxylase activity than xanthohumol, and also inhibited the Cyp1A2-mediated metabolism of aflatoxin B1, which suggests a high influence on the modulation of carcinogen metabolism [69,85,104].

### 6.2. Anti-Inflammatory and Antioxidant Properties

Beer volatiles and non-volatiles are extensively described as having antioxidant properties. It has been exhibited that xanthohumol inhibited LDL oxidation in vitro, induced by Cu^2+^, and reduced lipid peroxidation of liver microsomes in rats. Based on in vivo and in vitro studies it has also been reported that bio-compounds derived from humulones, cinnamic and benzoic acids, prenyl-chalcones procyanidins, and catechins are directly related to the high antioxidant capacity of beer. Finally, the anti-inflammatory effects of beer bioactive compounds are mainly related to the inhibition of both, cyclo-oxygenase 1 activity and the inducible nitric oxide synthase [24,85,92,105].

Oliveira Neto et al. (2017) investigated the antioxidant capacity of ale and lager beer samples by spectrophotometric and electroanalytical methods. It has been demonstrated that ale beers exhibit higher antioxidant activity than lager beer, mostly due to the higher fermentation temperature related to the brewing process of ale beer, leading to higher extraction of phenolic compounds. The identified substances appear to be only of natural sources, mainly from hops as isoxanthohumol, cohumulone, and humulone, from barley malt, as trihydroxyoctadecenoic acid (TOD), which is derived from linoleic acid and is formed during the brewing process. TOD is being described as responsible for beer flavor and microbiological stability and has antifungal activity. Catechin and caffeic acid can be from both, hops and barley, and also has proven great antioxidant activity [86,92].

Previously, the antioxidant capacity of xanthohumol has been investigated by scavenging 1,1-diphenyl-2-picrylhydrazyl (DPPH) radicals test, and the results showed that at a certain concentration (1 mM), xanthohumol exhibited a more efficient antioxidant activity than the reference compound (Trolox) in scavenging OH- and ROO- radicals. Moreover, xanthohumol showed an inhibition property on superoxide anion radical production in the xanthine/xanthine oxidase (X/XO) system and by TPA-stimulated HL-60 leukemia cells differentiated to granulocytes. Despite its mechanisms involved in the prevention of tumour promotion and antioxidant properties, xanthohumol also demonstrated the anti-inflammatory potential by inhibition of Cox-1 and -2 activities, and also prevented nitric oxide release by LPS-stimulated Raw 264.7 murine macrophages. Similar effects were described for additional hop components, and it is suggested that the inhibition is mediated at the protein level by the suppression of LPS-induced iNOS protein expression [85,92,105,106].

In a more recent study, Capece et al. (2018) evaluated the influence of a probiotic strain of *Saccharomyces cerevisiae* var. *boulardii* on the antioxidant and physicochemical profile of beers. By DPPH scavenging activity assays it was demonstrated that the inclusion of *Saccharomyces cerevisiae* var. *boulardii* in mixed starters (co-fermentations with *Saccharomyces cerevisiae*) led to an increase in the antioxidant activity in all the beers (an increase of ~135.68%), in comparison to values found for beer samples fermented with single yeast starters. As expected, the phenolic content was similar to the antioxidant activity and was significantly higher in beers from mixed starters (an increase of ~10.95%), indicating the influence of *Saccharomyces cerevisiae* var. *boulardii* strain on these parameters and a promising beneficial effect on human health. Capece et al. (2018) also evaluated the final content of the main volatile compounds and the results showed that the inclusion of *Saccharomyces cerevisiae* var. *boulardii* strain did not affect negatively beer organoleptic profile, which, according to the authors, is an important criterion that has to be analyzed before the use of probiotic yeast strains for food production [107].

### 6.3. Cardiovascular Diseases

Cardiovascular diseases are one of the main causes of death worldwide and require special attention from the scientific community (and researchers) in order to minimize their harmful effects. Phenolic compounds found in beer might play a major role in its beneficial effect on vascular diseases. However, studies have demonstrated that the ethanol itself might exert a protective effect on cardiovascular events when moderately consumed. It suggests that the protective effects on the cardiovascular system resulting from beer bio-compounds are mainly due to a combination of alcoholic and non-alcoholic constituents, with the polyphenolic compounds being the most well-described group [15,28,101,102,108].

Ethanol reduces cardiovascular risk factors, such as diabetes mellitus, reduced high-density lipoprotein (HDL) cholesterol levels, increased blood levels of low-density lipoprotein (LDL), cholesterol and triglycerides. Also, it is associated with the protection exerted on coronary heart disease and other cardiovascular outcomes, such as thrombosis and the fibrinolysis process. The most well-established mechanism to describe the ethanol protective effects against ischemic cardiovascular disease includes an increase in HDL cholesterol levels, reduction in platelet aggregation and in the levels of fibrinogen [15,85,101,102].

Different biological effects of polyphenolic concentrations in beer have been demonstrated by cell culture assays and enzyme experiments. Antioxidant, anti-inflammatory, anti-carcinogenic, estrogenic and even antiviral properties have been described to be associated with the moderate phenolic compounds, present in beer, intake. Different profiles of in vitro biological activity have been described for these compounds, which combined could have a synergistic vascular protective effect. Beneficial health effects of beer compounds against atherosclerosis development have also been demonstrated and are suggested to be due to polyphenolic activity on vascular function, as well as their antioxidant and anti-inflammatory properties [92,102,106].

Studies conducted by Chiva-Blanch et al. (2015), evaluated the effects of alcoholic and non-alcoholic components on atherosclerotic biomarkers in a randomized trial. It was suggested that the polyphenolic content of beer, similarly to wine polyphenols, reduces inflammatory biomarkers and leukocyte adhesion molecules, while its synergic effect with ethanol primarily improves the lipid profile and reduces some plasma inflammatory biomarkers related to atherosclerotic process [103].

De Gaetano et al.’s (2016) studies show a protective effect of beer compounds on the cardiovascular system, including ischemic stroke, congestive heart failure, peripheral arteriopathy, and coronary heart disease. As previously mentioned, polyphenols and their metabolites can contribute to protecting against cardiovascular diseases. Described polyphenols in beer and wine decreased blood pressure while increased plasma nitric oxide concentration, and, at a certain level, regulated the systolic and diastolic blood pressures. These collected data strengthen the idea of beer phenolic compounds as allies in the prevention of chronic and degenerative diseases [15,85,103].

## 7. Conclusions

Currently, several research groups are focusing their attention on the many perspectives that beer can bring to the scientific scenario. Since craft beers are becoming more popular and their consumption has increased in the past decade, it is important to know, study and characterize it, regarding their raw materials and chemical composition. In this review, it is possible to agree that esters are the major compound in terms of aroma profile and attest their biosynthetic pathway during the fermentation step is of main relevance for beer quality. Ethyl acetate is the ester with the highest concentration in beer, and its formation by brewer’s yeast is controlled mainly by the expression level of the AATase-encoding genes. In addition, changes in the availability of the two substrates for ester production, higher alcohols, and acyl-CoA, also influence ester synthesis rates. Thus, any factor that influences the expression of the ester synthase genes and/or the concentrations of substrates will affect ester production accordingly. Brewers, therefore, have a broad range of different ways at their disposal to control acetate ester production.

Beer is also described to be a major source of compounds that are associated with important health benefits. These beer compounds are found to significantly reduce the risks of cardiovascular disease development. Part of this protective effect of beer is due to their alcoholic content and in part to their non-alcoholic components, mainly polyphenols. The chemopreventive activities of some beer components have also been shown and are directly related to hop-derived prenylflavonoids and humulone. However, future clinical and in vivo studies concerning the bioavailability, distribution, and efficacy are still needed, since beer is a mixture of many compounds and their combination might act in synergy. Furthermore, future studies should also focus on how beer bio-compounds would elucidate their main bioactive functions for human health benefits in vivo.

## Figures and Tables

**Figure 1 molecules-24-01568-f001:**
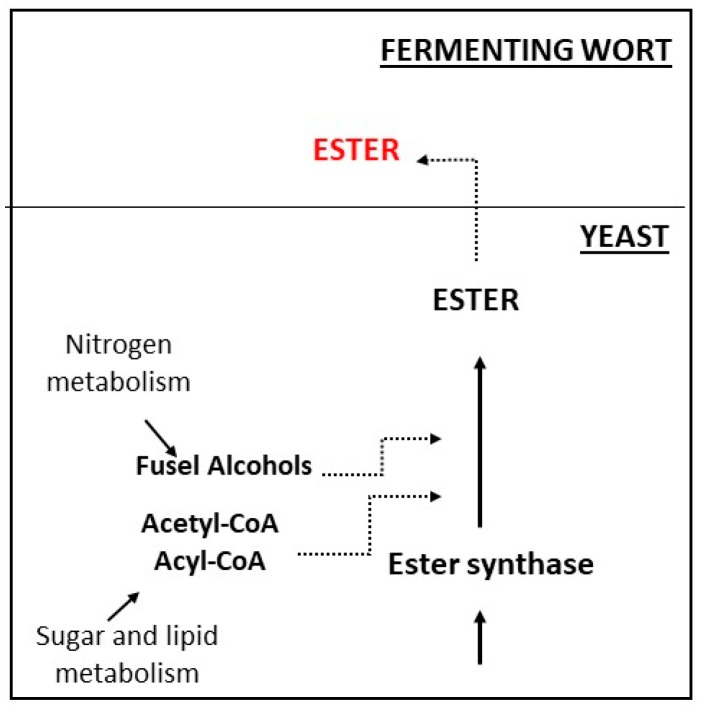
Ester synthesis pathway: During yeast metabolism, fermentable sugars and lipids are converted into acetyl-CoA and nitrogen metabolism induces the formation of fusel alcohols. These compounds are further used to produce ester molecules by the catalytic activity of the enzyme ester synthase. The picture was adapted from Verstrepen et al. (b) [39].

**Figure 2 molecules-24-01568-f002:**
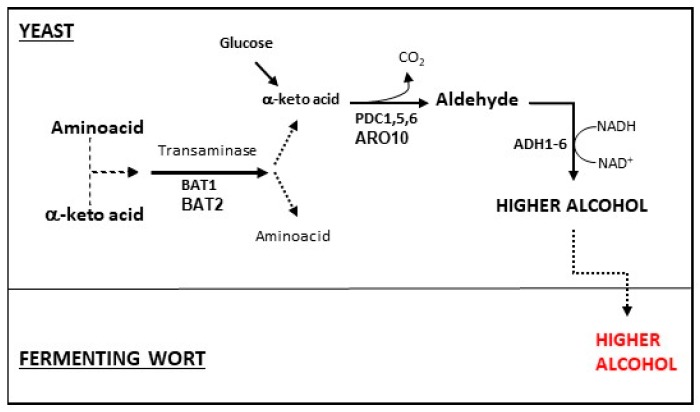
Higher alcohol formation pathways: In both pathways, from glucose and aminoacid, α-keto acids are formed, and further decarboxylated to produce aldehydes, and finally dehydrogenated to produce the corresponding primary alcohol. In the Ehrlich pathway or the aminoacids pathway, the aminoacids are converted to α-keto acids by a transaminase enzyme. The figure was adapted from Kobayashi et al. [9].

**Figure 3 molecules-24-01568-f003:**
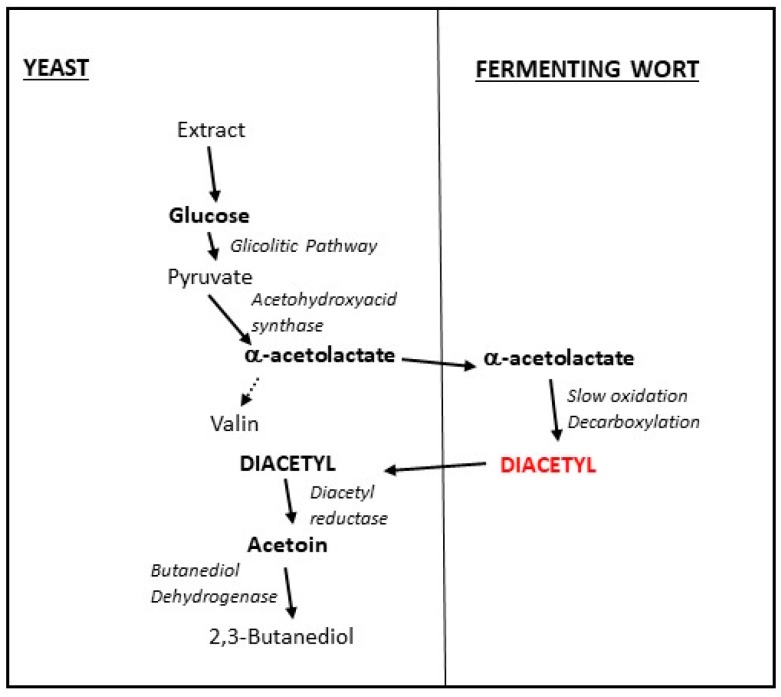
Diacetyl metabolism pathway: Yeast cells in the fermenting wort internalize the soluble sugars and activate the glycolytic pathway. Pyruvate is converted into α-acetolactate that is expelled from the cell and converted to diacetyl, which can remain in the final beer or be reabsorbed and converted into the non-flavour 2,3-butanediol. This figure was adapted from Kobayashi et al. (2005) [76].

**Table 1 molecules-24-01568-t001:** Beer flavor compounds in beer and their perception threshold well described in the literature.

Beer Compound	Concentration Found in Beer ^1^ (ppm)	Perception Threshold ^2^ (ppm)	References
Esters			
Ethyl acetate	15.3–16.8	5–10; 25–50	[16,39,40,64,65,66]
Phenyl ethyl acetate	0.1–0.73	3–5	[39,44,64,65,67]
Isoamyl acetate	0.078–0.489; 1.2	0.03; 1–2.5	[16,39,44,64,66]
Isobutyl acetate	0.03–1.2	0.5–1	[65,67]
Ethyl caproate (ethyl hexanoate)	0.081–0.411	0.014–0.2; 0.2–0.3	[16,39,44,64,66,67]
Ehtyloctanoate	0.04–0.53	0.9	[44,65,67]
Higher alcohol			
Amyl alcohol	8.73–44	50–70	[40,66,67]
Isobutyl alcohol	6.6; 58.9	100–175	[44,66,67]
Carbonyl compounds			
Acetaldehyde	0.952–8.1	1.114–5	[66,67,73]
Diacetyl	0.013–0.07	0.1–0.2	[16,66,67,73,74]

^1^ Flavor compound concentration may vary according to beer style; ^2^ Perception threshold values may vary according to beer style and body.

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
