# Peer review of "Beer Molecules and Its Sensory and Biological Properties: A Review"

_molecules, 2019, doi:10.3390/molecules24081568_

Round 1
Reviewer 1 Report
1. References throughout the text should be reviewed. Number and author styles are simultaneously used Ex: Line 405
2. There some ideas that are repeated. This should be eliminated. For example, the four well-known basic ingredients for beer production are presented several times in manuscript to explain and discuss some beer characteristics. In addition, the authors (singly) mention them again and again. This is generally known and should be eliminated.
§ Line 16: “beer is produced by the combination of water, barley malt, hops, and yeast”
§ Line 32: “water, barley malt, hops, and yeast still are the basic raw materials for brewing”
§ Line 45: “The beer aroma is known to be derived primarily from innate chemical volatile compounds of the barley malt (or as a result of the thermal treatment during malting), hops and yeast metabolism”
§ Line 101: “Beers are mostly brewed from the four primary ingredients: water, barley malt, hops and yeast”
3. The comment 2) also extends to the different types of beer: ale and lager (the idea on line 92-96 has been presented in lines 40-44). This type of information is common knowledge and duplication should be avoided.
4. Please provide additional details about the ratio stated at line 287.
5. Authors state that diacetyl concentration should not exceed the 5ppm (Line 320). This is about 25-50 times above the diacetyl odour threshold. Please explain the 5 ppm limits.
6. Diacetyl is of the most critical compounds in beer flavor. I consider that additional information regarding their sensory impact in ale and lager beer should be introduced.
7. Review some typos in text, for example:
§ Line 336; Line 363; line394;
8. I have some concerns regarding “beneficial health effects attributed to beer consumption” (line 395). Please explain and detailed the studies that prove this. I consider that these kinds of statements require specific studies (example: clinical trial, or in vivo animal trial with a designated biomarker of human health, …) and should not be concluded from merely beer chemical characterization. In this regard, I suggest that authors review the manuscript, in particular section 6, and explain in which facts that conclusions arise.
Author Response
Manuscript ID: molecules-472102
Answer to the comments - Reviewer #1
Reviewer: 1. References throughout the text should be reviewed. Number and author styles are simultaneously used Ex: Line 405
Answer: As recommended by the reviewer, the references throughout the text were corrected in the revised version of the manuscript.
Reviewer: 2. There some ideas that are repeated. This should be eliminated. For example, the four well-known basic ingredients for beer production are presented several times in manuscript to explain and discuss some beer characteristics. In addition, the authors (singly) mention them again and again. This is generally known and should be eliminated.
Answer: All mistakes pointed out by the Reviewer were corrected in the new version.
Abstract:
Reviewer: Line 16: “beer is produced by the combination of water, barley malt, hops, and yeast”
Answer: The mistake was fixed.
Reviewer: Line 32: “water, barley malt, hops, and yeast still are the basic raw materials for brewing”
Answer: The mistake was fixed.
Reviewer: Line 45: “The beer aroma is known to be derived primarily from innate chemical volatile compounds of the barley malt (or as a result of the thermal treatment during malting), hops and yeast metabolism”
Answer: The mistake was fixed.
Reviewer: Line 101: “Beers are mostly brewed from the four primary ingredients: water, barley malt, hops and yeast”
Answer: The mistake was fixed.
Reviewer: 3. The comment 2) also extends to the different types of beer: ale and lager (the idea on line 92-96 has been presented in lines 40-44). This type of information is common knowledge and duplication should be avoided.
Answer: Following the recommendation of the Referee, mistakes were corrected the revised version of the manuscript.
Reviewer: 4. Please provide additional details about the ratio stated at line 287.
Answer: As recommended by the reviewer, Additional details about the inoculation ratio were provided and the whole sentence was rewritten in the new version of the manuscript.
Reviewer: 5. Authors state that diacetyl concentration should not exceed the 5ppm (Line 320). This is about 25-50 times above the diacetyl odour threshold. Please explain the 5 ppm limits.
Answer: The diacetyl concentration value was first typed by mistake, and the authors did not realize before paper submission. The correct value was inserted in the revised version of the manuscript.
Reviewer: 6. Diacetyl is of the most critical compounds in beer flavor. I consider that additional information regarding their sensory impact in ale and lager beer should be introduced.
Answer: Additional information regarding sensory impact of diacetyl was incorporated in the new version of the manuscript as suggested by the reviewer.
Reviewer: 7. Review some typos in text, for example:
Reviewer: Line 336; Line 363; line394;
Answer: The mistake was fixed.
Reviewer: 8. I have some concerns regarding “beneficial health effects attributed to beer consumption” (line 395). Please explain and detailed the studies that prove this. I consider that these kinds of statements require specific studies (example: clinical trial, or in vivo animal trial with a designated biomarker of human health, …) and should not be concluded from merely beer chemical characterization. In this regard, I suggest that authors review the manuscript, in particular section 6, and explain in which facts that conclusions arise.
Answer: As stated by the reviewer, the “beneficial health effects attributed to beer consumption” (Line 416) is a controversial discussion and requires more specific studies, so the authors decided to avoid the discussion concerning beer consumption and focused on the individual health benefits of each group of compounds found in beer.
Finally, we would like to thank the comments made by the Reviewer 1, and we believe that the actual version of the manuscript can be useful to the field of natural products chemistry.

Reviewer 2 Report
This review is aimed to describe the synthesis and role of beer compounds under different aspects, in particular on chemical, sensorial and health benefits. The review reports numerous data from very recent papers, representing an interesting update on the recent findings in compounds playing an important role on beer quality. As a consequence, I consider the manuscript suitable for the publication on Molecules after minor revision.
The following points have be considered before the publication:
- Paragraph 4, lines 241-245: this part is an introduction to the different classes of compounds affecting the final flavor quality of beer; by my opinion, this part has to be moved before this point, i.e. before the paragraph 3.
- Line 307: Saccharomyces pombe, this species is not present at the date; probably, it was a mistake, the species has to be Schizosaccharomyces pombe.
- Line 336: a space between “of” and “fermentation” is necessary.
- Line 363: “I” has to be changed into “It”.
- A table summarizing the main compounds affecting beer aroma, concentration usually present in the beer e threshold values might be useful to improve the quality of paper.
- As regards the influence of beer compounds on human health (paragraph 6), I suggest to report the recent publications related to the influence of yeast starter on antioxidant acitivty of beer (i.e. Capece et al. 2018. Int. J. Food Microbiol. 284, 22–30.)
- A revision of English language is necessary. For example, the following sentences have to be checked:
- Abstract:
- lines 16-19: During brewing, fermentation step many volatile and phenolic 16 compounds are produced and bring singular organoleptic characteristics……;
- Lines 20-22: …..since it is when 20 alcohol and many volatiles esters are produced and metabolized by the yeast, and therefore 21 demands a strict control.
Author Response
Manuscript ID: molecules-472102
Answer to the comments - Reviewer #2
Reviewer: This review is aimed to describe the synthesis and role of beer compounds under different aspects, in particular on chemical, sensorial and health benefits. The review reports numerous data from very recent papers, representing an interesting update on the recent findings in compounds playing an important role on beer quality. As a consequence, I consider the manuscript suitable for the publication on Molecules after minor revision.
The following points have be considered before the publication:
Reviewer: Paragraph 4, lines 241-245: this part is an introduction to the different classes of compounds affecting the final flavor quality of beer; by my opinion, this part has to be moved before this point, i.e. before the paragraph 3.
Answer: As suggested by the reviewer, was changed the lines 255-259 of paragraph 4 to paragraph 1 (lines 49-53).
Reviewer: Line 307: Saccharomyces pombe, this species is not present at the date; probably, it was a mistake, the species has to be Schizosaccharomyces pombe.
Answer: The mistake was fixed.
Reviewer: Line 336: a space between “of” and “fermentation” is necessary.
Answer: The mistake was fixed.
Reviewer: Line 363: “I” has to be changed into “It”.
Answer: The mistake was fixed.
Reviewer: A table summarizing the main compounds affecting beer aroma, concentration usually present in the beer e threshold values might be useful to improve the quality of paper.
Answer: The table summarizing the main compounds affecting beer flavor and aroma, concentrations and threshold values was inserted in the revised version of the manuscript.
Reviewer: As regards the influence of beer compounds on human health (paragraph 6), I suggest to report the recent publications related to the influence of yeast starter on antioxidant activity of beer (i.e. Capece et al. 2018. Int. J. Food Microbiol. 284, 22–30.)
Answer: As suggested by the reviewer, was mentioned publication related to the influence of yeast starter on antioxidant activity of beer in the new version of the manuscript. Moreover, we thank you for the great suggestion.
Reviewer: A revision of English language is necessary. For example, the following sentences have to be checked:
Answer: Following the recommendation of the Referee, we have checked the English of the revised version of the manuscript.
Abstract:
Reviewer: lines 16-19: During brewing, fermentation step many volatile and phenolic 16 compounds are produced and bring singular organoleptic characteristics……;
Answer: The mistake was fixed.
Reviewer: Lines 20-22: …..since it is when 20 alcohol and many volatiles esters are produced and metabolized by the yeast, and therefore 21 demands a strict control.
Answer: The mistake was fixed.
Finally, we would like to thank the comments made by the Reviewer 2, and we believe that the actual version of the manuscript can be useful to the field of natural products chemistry.

Reviewer 3 Report
I have only reviewed the first two sections of this document (up to line 110). This has taken me over two hours! The syntax of the manuscript requires detailed and complete revision.
Author Response
Manuscript ID: molecules-472102
Answer to the comments - Reviewer #3
Reviewer: I have only reviewed the first two sections of this document (up to line 110). This has taken me over two hours! The syntax of the manuscript requires detailed and complete revision.
Answer: Following the recommendation of the Referee, all mistakes pointed out were corrected by the authors and by a native English-speaking colleague in the new version of the manuscript.
Finally, we would like to thank the comments made by the Reviewer 3, and we believe that the actual version of the manuscript can be useful to the field of natural products chemistry.

Round 2
Reviewer 3 Report
There are a large number of changes suggested. Consequently, each page has been scanned. Please check that the cited reference numbers in the text correspond to the references listed in the reference section of the manuscript.

Author Response
Manuscript ID: molecules-472102
Answer to the comments - Reviewer #3
Reviewer: There are a large number of changes suggested. Consequently, each page has been scanned. Please check that the cited reference numbers in the text correspond to the references listed in the reference section of the manuscript.
Answer: We agree with the reviewer and incorporated all suggestions according to each scanned page. As recommended by the reviewer, we also checked all references in the revised version of the manuscript.
Finally, we would like to thank the comments made by the Reviewer 3, and we believe that the actual version of the manuscript can be useful to the field of natural products chemistry.
